# Synthesis and Properties of Pentafluorosulfanyl Group (SF_5_)-Containing Meta-Diamide Insecticides

**DOI:** 10.3390/molecules25235536

**Published:** 2020-11-25

**Authors:** Jae Gon Kim, On-Yu Kang, Sang Mee Kim, Guldana Issabayeva, In Seok Oh, Yaeji Lee, Won Hyung Lee, Hwan Jung Lim, Seong Jun Park

**Affiliations:** 1Bio & Drug Discovery Division, Korea Research Institute of Chemical Technology (KRICT), 141 Gajeong-ro, Yuseong-gu, Daejeon 34114, Korea; jgkim@krict.re.kr (J.G.K.); dhdb0901@krict.re.kr (O.-Y.K.); smk95@krict.re.kr (S.M.K.); guldana@krict.re.kr (G.I.); ois0821@krict.re.kr (I.S.O.); leeyj@krict.re.kr (Y.L.); 2Department of Chemistry, Sungkyunkwan University, Suwon 16419, Korea; 3Department of Medicinal and Pharmaceutical Chemistry, University of Science & Technology, Daejeon 34113, Korea; 4Department of Chemistry, Sogang University, Seoul 04107, Korea; 5Central Research Institute, Kyung Nong Co. Ltd., 34-14 Summeori-gil, Kyongju 38175, Korea; whlee1@dongoh.co.kr

**Keywords:** meta-diamide, pentafluorosulfanyl, insecticide, GABARs

## Abstract

Herein, we describe novel pentafluorosulfanyl (SF_5_) group-containing meta-diamide insecticides. For the facile preparation of the SF_5_-based compounds **4a**–**d**, practical synthetic methods were applied. Among newly synthesized compounds, 3-benzamido-*N*-(2,6-dimethyl-4-(pentafluoro-λ^6^-sulfanyl)phenyl)-2-fluorobenzamide **4d** showed (i) a high insecticidal activity, (ii) an excellent selectivity to insects, and (iii) good levels of water solubility and log P values. In this study, we demonstrated that the pentafluorosulfanyl moiety could serve as an attractive functionality for the discovery of a new scope of crop-protecting agents.

## 1. Introduction

The introduction of a fluorine atom into a biologically active compound can have a significant influence on its properties. One or more incorporated fluorine atoms can alter the electrostatic and hydrogen bonding parameters of the molecule as well as its physicochemical and pharmacokinetic properties [1,2]. Currently, fluorine-containing substituents, which are commonly encountered in commercial pharmaceuticals and agrochemicals, include fluoroaromatic, trifluoromethyl (CF_3_), trifluoromethoxy (OCF_3_), and trifluoromethlythio (SCF_3_) functionalities [3,4]. Another fluorinated substituent that reflects the continuing development of a relatively new fluorinated building block with distinct properties could be the pentafluorosulfanyl (SF_5_) group. The SF_5_ group is often called the “super-trifluoromethyl group”, and aryl sulfanyl pentafluorides display high thermal and chemical stability, electronegativity, and lipophilicity [5,6,7]. Due to its unique properties, the SF_5_ group has widely been applied in drug discovery and crop protection research (Figure 1) [8,9,10,11,12,13,14], since the first organic pentafluorosulfanyl compound was described in 1950 [15].

As a part of our ongoing efforts to discover new eco-friendly insecticides [16,17,18], we are particularly interested in small molecules containing the fluorine atom. One of the outstanding representatives of this category is broflanilide, which is known as an efficient broad-spectrum meta-diamide insecticide containing a high number of fluorine atoms [19,20]. Taking into account the suggested potential bioisosteric relationship between the SF_5_ and CF_3_ substituents [2,5], we proposed the novel design of the SF_5_-containing meta-diamide insecticide **4d** (Figure 2). In order to examine the influence of SF_5_ moiety on the properties of the meta-diamide **4d** (R^1^ = H, R^2^ and R^3^ = CH_3_) and investigate the significance of the effect caused by the replacement of the CF(CF_3_)_2_ group to SF_5_ functionality, the known meta-diamide insecticide **BPB1** (R^1^ = H, R^2^ and R^3^ = CH_3_) was selected for this study (Figure 2). According to the previous studies on the development of meta-diamide-based insecticides, it was discovered that the insecticide **BPB3** (R^1^ = CH_3_) can be metabolized to its active form, **BPB1** (R^1^ = H), which in its turn demonstrates insecticidal activity by acting on the target RDL GABARs gene subunit and inhibiting its expression [19,20].

## 2. Results

The synthetic route shown in Scheme 1 was successfully applied for the preparation of the *para*-SF_5_ substituted aniline derivatives, **1b**–**d**. The synthesis was initiated by the bromination of a commercially available aniline, **1a** using *N*-bromosuccinimide (NBS), which led to the formation of mono- and di-bromo anilines **1a′** and **1a″**, respectively. The molecules, **1a′** and **1a″** were subsequently methylated by the Pd-catalyzed cross-coupling to give the corresponding anilines, **1b** and **1d** in 64% and 75% yields, respectively. Finally, 2-methyl-aniline **1b** was further reacted with NBS in DMF to produce 2-methyl-6-bromo-aniline **1c** with excellent yield [21].

The target compounds with the incorporated SF_5_ group were successfully prepared starting from 2-fluoro-3-nitrobenzoic acid, which is commercially available and can be readily converted into the corresponding aniline **2 [22]**. Benzoylation of **2** provided 3-benzamido-2-fluorobenzoic acid **3** in excellent yield [23]. Then, SF_5_-containing compounds **4a**–**d** were easily prepared by the condensation of benzoic acid **3** with 4-SF_5_-anilines **1a**–**d** (Scheme 2) [24].

The synthesized SF_5_-containing derivatives **4a**–**d** were examined for their insecticidal activities at 10 ppm concentration against the 3rd instar larvae of *Plutella xylostella* using the leaf-dip method [16,17,18,25]. Among them, the compounds **4c** and **4d** showed excellent activities with high inhibition of feeding behaviors (entry 3 and 4, Table 1). Interestingly, 2,6-dimethyl-substituted compound **4d**, SF_5_-containing meta-diamide **BPB1**, displayed an excellent potency with eating area—0~5%. According to the data in Table 1, it is reasonable to believe that the SF_5_ group can be considered as an important part of the toxophore.

The target site specificities of newly prepared SF_5_-based meta-diamide insecticide **4d** should differ in insect and mammalian GABA and glycine receptors. In this regard, the cell-based antagonist activities of the meta-diamide **4d** against the human GABA_A_R and glycine receptor (GlyR) A1 were investigated. According to the results obtained from previous studies, the mammalian GABA_A_R α1β3γ2 and the human glycine receptor (GlyR) A1 were selected for this study [19,26,27]. As shown in Table 2, the estimated IC_50_ values of SF_5_-containing meta-diamide **4d** and broflanilide against GABA_A_R and GlyR were more than 30 μM. This discovery implies that SF_5_-containing meta-diamide **4d** has much higher selectivity toward targeted insects.

There are a number of studies for confirming the existence of the strong relationship between insecticidal activity and bioavailability of the potential insecticides [18,28]. In this context, SF_5_-containing compound **4d**, which showed the highest potency, was further investigated in its physicochemical properties, including LogP and solubility. As a reference, properties of broflanilide were also measured. For lipophilicity, most commonly referred to as LogP [29], replacing the heptafluoroisopropyl group with the SF_5_ moiety resulted in similar LogP values in broflanilide and the meta-diamide **4d** (entry 1, Table 3). In addition, both the molecules meta-diamide **4d** and broflanilide showed high levels of kinetic solubility [30].

Generally, the presence of fluoroaromatics and perfluoroalkanes increases the lipophilicity values of the molecules in comparison to the parental hydrocarbon bonds [31,32,33,34,35]. In addition to that, regarding its structural differences, the SF_5_ group possesses superior properties over other available fluorine-containing functionalities. Taking into consideration that lipophilicity plays a key role in transport processes [36], this result could be an important finding to discover new functionalities for application in the development of crop protecting agents.

## 3. Material and Methods

### 3.1. General Information

Melting points: Barnstead/Electrothermal 9300—measurements were performed in open glass capillaries. NMR spectra: Bruker AV 300 MHz (Bruker corporation, Billerica, MA, USA)(^1^H-NMR: 300 MHz, ^13^C-NMR: 75 MHz), AV 400 MHz (^1^H-NMR: 400 MHz, ^13^C-NMR: 100 MHz), AV 500 MHz (^1^H-NMR: 500 MHz, ^13^C-NMR: 125 MHz), and AV2 500 MHz (^19^F-NMR: 470 MHz); the spectra were recorded in CDCl_3_ and DMSO-d_6_ using tetramethylsilane (TMS) as the internal standard and are reported in ppm. 1H-NMR data are reported as follows: (s—singlet; d—doublet; t—triplet; q—quartet; dd—doublet of doublet; m—multiplet; coupling constant(s) J are given in Hz; integration, proton assignment). High-resolution mass spectra (HRMS): JEOL JMS-700.

*2-Methyl-4-(pentafluorothio)aniline* (**1b**) [37,38]. A mixture of 2-bromo-4-(pentafluorothio)aniline (500 mg, 3.223 mmol), methylboronic acid (2.0 eq., 6.446 mmol), Pd(dppf)Cl_2_.DCM (0.1 eq., 0.322 mmol), and Cs_2_CO_3_ (3.0 eq., 9.669 mmol) in 1,4-dioxane (8.6 mL) was stirred at 105 °C for 5 h. The reaction mixture was diluted with EtOAc and washed with aq. NaHCO_3_. The organic layer was dried over Na_2_SO_4_ and concentrated. The residue was purified by flash column chromatography on a silica gel (hexane:EtOAc = 15:1) to give the desired product **1b** as a yellow solid (481 mg, 64%).

*2-bromo-6-methyl-4-(pentafluorothio)aniline* (**1c**). To a solution of 2-methyl-4-(pentafluorothio)aniline (350 mg, 1.5 mmol) in DMF (15 mL), NBS (1.03 eq., 1.545 mmol) was added. The reaction mixture was stirred at RT for 2 h, quenched with water, and extracted with EtOAc (10 mL). The organic layer was dried over NaSO_4_, filtered, and concentrated. The residue was purified by column chromatography on a silica gel (hexane:EtOAc = 20:1) to give the desired product **1c** as a red solid (459 mg, 98%). mp 64~65 °C; ^1^H-NMR (500 MHz, CDCl_3_) δ 7.71 (d, *J* = 2.5 Hz, 1H), 7.40 (d, *J* = 2.5 Hz, 1H), 4.42 (s, 2H), 2.25 (s, 3H); ^13^C-NMR (100 MHz, CDCl_3_) δ 146.0, 144.7, 129.2, 127.9, 122.8, 107.8, 19.4; ^19^F-NMR (470 MHz, CDCl_3_) δ 86.46 (p, 1F, *J*_SF–SF4_ = 150.3 Hz, SF), 64.90 (d, 4F, *J*_SF4–SF_ = 150.2 Hz, SF_4_); HRMS (EI) calcd. for C_7_H_7_BrF_5_NS 310.9403, found 310.9409 (see Appendix A).

*2,6-dimethyl-4-(pentafluorothio)aniline* (**1d**). A mixture of 2-bromo-6-methyl-4-(pentafluorothio)aniline (200 mg, 0.64 mmol), methylboronic acid (2.0 eq., 1.28 mmol), Pd(dppf)Cl_2_.DCM (0.1 eq., 0.064 mmol), and Cs_2_CO_3_ (3.0 eq., 1.92 mmol) in 1,4-dioxane (1.7 mL) was stirred at 105 °C for 5 h. The mixture was diluted in EtOAc, washed with aq. NaHCO_3_, dried over Na_2_SO_4_, and concentrated. The residue was purified by flash column chromatography on a silica gel (hexane:EtOAc = 15:1) to give the desired product **1d** as a brown solid (119 mg, 75%). mp 205~206 °C; ^1^H-NMR (300 MHz, CDCl_3_) δ 7.34 (s, 2H), 3.90 (s, 2H), 2.20 (s, 6H); ^13^C-NMR (100 MHz, CDCl3) δ 145.2, 144.1, 128.0, 126.5, 121.8, 120.7, 18.0; ^19^F-NMR (470 MHz, CDCl_3_) δ 87.32 (p, 1F, *J*_SF–SF4_ = 149.9 Hz, SF), 64.88 (d, 4F, *J*_SF4–SF_ = 149.8 Hz, SF_4_); HRMS (EI) calcd. for C_8_H_10_F_5_NS 247.0454, found 247.0451 (see Appendix A).

*3-benzamido-2-fluorobenzoic acid* (**3**). To 2-fluoro-3-nitro-benzoic acid (2 g, 10.8 mmol) in 44 mL of tetrahydrofuran, 20% palladium hydroxide on carbon (148 mg, 1.05 mmol) was added. The reaction was stirred under hydrogen for 2 h. The reaction mixture was filtered through a short pad of celite and the solution was evaporated (without a purification) to give the desired compound **3** as an ivory color solid (1.64 g, 98%). mp 257~258 °C; ^1^H-NMR (300 MHz, DMSO-d_6_) δ 13.32 (s, 1H), 10.22 (s, 1H), 8.02–7.95 (m, 2H), 7.86–7.79 (m, 1H), 7.76–7.69 (m, 1H), 7.66– 7.59 (m, 1H), 7.58–7.51 (m, 2H), 7.31 (t, *J* = 7.8 Hz, 1H); ^13^C-NMR (100 MHz, DMSO-d_6_) δ 165.5, 164.8, 133.7, 131.9, 131.1, 128.5, 128.5, 128.3, 127.8, 126.9, 123.8, 123.8; ^19^F-NMR (470 MHz, CDCl_3_) δ −119.6; HRMS (EI) calcd. for C_14_H_10_FNO_3_ 259.0645, found 259.0638 (see Appendix A).

### 3.2. General Method for the Synthesis of **4a**–**d**

A mixture of 3-benzamido-2-fluorobenzoic acid **3** (50 mg, 0.193 mmol) and SOCl_2_ (3.0 eq., 0.579 mmol) was refluxed for 2 h. The solution of aniline (0.9 eq., 0.174 mmol) and NaHCO_3_ (2.7 eq., 0.52 mmol) in acetone/water (0.4 mL/0.04 mL) was added in the reaction mixture. The reaction mixture was refluxed for 1 h, quenched with water, and extracted with EtOAc (10 mL). The organic layer was dried over NaSO_4_, filtered, and concentrated. The residue was purified by column chromatography on a silica gel (hexane:EtOAc = 10:1) to give the desired product.

*3-Benzamido-2-fluoro-N-(4-(pentafluorothio)phenyl)benzamide* (**4a**). This follows the general method. The residue was purified by column chromatography on a silica gel (Hexane:EtOAc = 10:1) to give the desired diamide **4a** as a white solid (55.3 mg, 78% yield). mp 193~194 °C; ^1^H-NMR (500 MHz, CDCl_3_) δ 8.63–8.57 (m, 1H), 8.44 (d, *J* = 12.4 Hz, 1H), 8.09 (s, 1H), 7.94–7.88 (m, 2H), 7.85–7.80 (m, 1H), 7.78 (s, 4H), 7.65–7.60 (m, 1H), 7.57–7.52 (m, 2H), 7.36 (t, *J* = 8.0 Hz, 1H); ^13^C-NMR (100 MHz, CDCl_3_) δ 165.6, 161.3, 140.2, 133.9, 132.6, 129.1, 127.2, 127.1,127.0, 126.8, 126.4, 126.3, 125.5, 125.4, 121.3, 121.2, 119.6; ^19^F-NMR (470 MHz, CDCl_3_) δ 84.83 (quin, 1F, *J*_SF–SF4_, *J* = 150.3 Hz, SF), 63.41 (d, 4F, *J*_SF4–SF_, *J* = 149.8 Hz, SF_4_), −131.28–−131.40 (m, 1F); HRMS (EI) calcd. for C_20_H_14_F_6_N_2_O_2_S 460.0680, found 460.0680 (see Appendix A).

*3-benzamido-2-fluoro-N-(2-methyl-4-(pentafluorothio)phenyl)benzamide* (**4b**). This follows the general method. The residue was purified by column chromatography on a silica gel (Hexane:EtOAc = 10:1) to give the desired diamide **4b** as a white solid (54.3 mg, 73% yield). mp 189~190 °C; ^1^H-NMR (500 MHz, CDCl_3_) δ 8.64–8.59 (m, 1H), 8.42–8.34 (m, 2H), 8.11 (s, 1H), 7.94–7.87 (m, 3H), 7.67 (dd, *J* = 9.0, 2.7 Hz, 1H), 7.64–7.60 (m, 2H), 7.56–7.52 (m, 2H), 7.38 (t, *J* = 8.0 Hz, 1H), 2.42 (s, 3H); ^13^C-NMR (100 MHz, CDCl_3_) δ 165.6, 160.9, 152.2, 149.8, 138.6, 134.0, 132.6, 129.0, 127.9, 127.6, 127.1, 126.9, 126.8, 126.72, 126.70, 126.46, 126.44, 125.49, 125.46, 125.0, 121.3, 121.1, 121.0, 18.0; ^19^F-NMR (470 MHz, CDCl_3_) δ 85.10 (t, 1F, *J*_SF–SF4_, *J* = 150.2 Hz, SF), 63.42 (d, 4F, *J*_SF4–SF_, *J* = 149.9 Hz, SF_4_), −132.30 (s, 1F); HRMS (EI) calcd. for C_21_H_16_F_6_N_2_O_2_S 474.0837, found 474.0837 (see Appendix A).

*3-benzamido-N-(2-bromo-6-methyl-4-(pentafluorothio)phenyl)-2-fluorobenzamide* (**4c**). This follows the general method. The residue was purified by column chromatography on a silica gel (Hexane:EtOAc = 10:1) to give the desired diamide **4c** as a brown solid (51.2 mg, 54% yield). mp 209~210 °C; ^1^H-NMR (500 MHz, CDCl_3_) δ 8.69–8.63 (m, 1H), 8.17–8.07 (m, 2H), 7.94–7.89 (m, 3H), 7.89–7.83 (m, 1H), 7.67 (d, *J* = 2.5 Hz, 1H), 7.64–7.59 (m, 1H), 7.56–7.52 (m, 2H), 7.40–7.35 (m, 1H), 2.43 (s, 3H); ^13^C-NMR (100 MHz, CDCl_3_) δ 165.6, 161.0, 152.4, 149.9, 138.9, 137.0, 134.0, 132.6, 129.1, 128.0, 127.6, 127.2, 127.1, 127.0, 126.5, 126.3, 125.4, 125.3, 121.4, 120.5, 120.4, 20.0; ^19^F-NMR (470 MHz, CDCl_3_) δ 82.97 (quin, 1F, *J*_SF–SF4_, *J* = 150.5 Hz, SF), 63.35 (d, 4F, *J*_SF4–SF_, *J* = 150.0 Hz, SF_4_), −130.98 (s, 1F); HRMS (EI) calcd. for C_21_H_15_BrF_6_N_2_O_2_S 551.9942, found 551.9954 (see Appendix A).

*3-benzamido-N-(2,6-dimethyl-4-(pentafluorothio)phenyl)-2-fluorobenzamide* (**4d**). This follows the general method. The residue was purified by column chromatography on a silica gel (Hexane:EtOAc = 10:1) to give the desired diamide **4d** as a white solid (52.7 mg, 70% yield). mp 205~206 °C; ^1^H-NMR (500 MHz, CDCl_3_) δ 8.60 (t, *J* = 7.9 Hz, 1H), 8.13 (s, 1H), 7.93–7.90 (m, 2H), 7.85–7.80 (m, 2H), 7.64–7.59 (m, 1H), 7.56–7.51 (m, 4H), 7.36 (t, *J* = 8.0 Hz, 1H), 2.36 (s, 6H); ^13^C-NMR (100 MHz, CDCl_3_) δ 165.6, 161.4, 161.3, 136.5, 136.4, 133.9, 132.5, 129.0, 127.1, 126.9, 126.8, 126.4, 126.4, 126.1, 126.15, 125.8, 125.8, 125.7, 125.3, 125.2, 18.9; ^19^F-NMR (470 MHz, CDCl_3_) δ 84.69 (quin, 1F, *J*_SF–SF4_, *J* = 150.3, SF), 63.09 (d, 4F, *J*_SF4–SF_, *J* = 149.7 Hz, SF_4_), −131.55 (s, 1F); HRMS (EI) calcd. for C_22_H_18_F_6_N_2_O_2_S 488.0993, found 488.0988 (see Appendix A).

## 4. Conclusions

In summary, starting from the known meta-diamide **BPB1** containing a heptafluoroisopropyl group and its isosteric replacement with pentafluorosulfanyl moiety (-SF_5_) led to the meta-diamide insecticide **4d**, a compound with good potency, high selectivity toward insects, and a similar level of lipophilicity with broflanilide. For the preparation of SF_5_-containing meta-diamide insecticides **4a–d**, an efficient synthetic route was established. This study has demonstrated that the pentafluorosulfanyl group (-SF_5_) could be an appealing structural scaffold for the discovery of a new crop-protecting agent.

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
