# Peer review of "Synthesis and Properties of Pentafluorosulfanyl Group (SF5)-Containing Meta-Diamide Insecticides"

_molecules, 2020, doi:10.3390/molecules25235536_

Round 1

Reviewer 1 Report

The general idea of this paper is about using of SF5-group to replace commongly exploited fluorine-containing substituents, like CF3 and C3F7. The authors present a reasonable synthetic routes to the target molecules, which, as it was designed, show the desired crop-protecting properties.

Interesting - to my opinion - is the fact that the presence of R1=Me in Broflanilide is not necessary for the biological activity requested. Why am I interested in this question - since the presence of two bulky groups at nitrogen atom would force another conformation of the amidic fragment to occur. From the results shown it's clear that the nature of the "left" part of the molecule is not so much significant. But this is a field for the further investigations.

 The paper could be published as it is

Author Response

We would really appreciate your comments. It will be very helpful for our researches in the future.

Reviewer 2 Report

Professors Lim and Park reported the synthesis of new compounds containing a Pentafluorosulfonyl group. The meta diamides synthesised have been tested in several biological applications. The results obtained are similar to the Broflanilide, showing that SF5 can substitue (CF3)CF group efficiently. 

The synthesis is correct and the Experimental Procedure looks correct and the SI is adequate

In my opinion, the study present enough novelty and interest to be published in Molecules, therefore I recommend its publication. 

Author Response

(The authors gave the same response as above.)

Reviewer 3 Report

Presented material brings to the light a specific substitution of biologically active derivatives that have been in addition modified by incorporation of fluoride atoms. The quality of obtained molecules has been properly supported by the analytical tools and the interpretation of the data is adequate. It is rather incremental work but important while thinking about introduction of new molecules active as pesticides/insecticides. It is an important field of research to face the problems of modern world. The presented molecules showed a behaviour promissing from that perspective opening further options for exploration. I recommend acceptance in current form. 

Author Response

(The authors gave the same response as above.)

Reviewer 4 Report

Generally, this is an interesting manuscript. The authors describe syntheses of SF5 analogues of the insecticide “Broflanilide” using known simple organic chemistry transformation (e.g. Schotten-Baumann amidation as the key-step) and insecticidal activity of the new compounds in comparison to the lead compound. The manuscript is well written, the experiments are clearly described and the structure of the products were doubtlessly assigned using the common spectroscopic and spectrometric methods. Copies of the NMR spectra are presented in the Supporting Information demonstrating the purity of the products. My main concern is whether Molecules is an appropriate journal for the presented results. The most interesting part of the manuscript is the insecticidal potency of the compounds, which is approximately the same as that of the lead compound. In my opinion, a more specific journal positioned to pesticides, agriculture or bioorganic chemistry might be more appropriate to presented these results.

Specific comments:

In my opinion the title of the manuscript is too general, not clear enough and might even be misleading. The authors describe synthesis and insecticidal activity of pentafluorosulfanyl group-containing N-benzoyl-3-aminobenoic acid amides. The term ”meta-diamides” is meaningless with regard to the structure of products and should be eliminated from the whole manuscript.

In the abstract, the term “good levels of physicochemical properties” is too general and meaningless. Why not denominate that water solubility and log P values were measured.

Page 2: authors might want to add an information on the steric requirements of the SF5 group in relation to the CF(CF3)2 group. I doubt that the SF5 group is an isosteric replacement as stated in the Conclusions. This group might be electronically similar (group electronegativity, electron withdrawing effect) and comparable regarding the influence on lipophilicity. Regarding the physicochemical and biological effects of fluorine and polyfluorinated groups, I like to draw the attention of authors on three recent reviews by Muller, by Pertusiati et al. and by Ojima et al. published in the book “Fluorine in Life Sciences”, Academic Press, 2019, pp 91-139. Pp 141-180 and pp 181-211. This should also be considered at page 4, last paragraph and in ref. [31].

Page 2: it is not clear whether compounds 1b-1d are known compounds described in the patent referenced in [24] already.

Finally, authors might want to comment on whether the SF5 derivatives might compete with the CF(CF3)2 lead in terms of economy and persistence in nature.

Author Response

We would really appreciate your comments. It will be very helpful for our researches in the future.

For your specific comments, we would like to address in most points.

The authors describe synthesis and insecticidal activity of pentafluorosulfanyl group-containing N-benzoyl-3-aminobenoic acid amides. The term ”meta-diamides” is meaningless with regard to the structure of products and should be eliminated from the whole manuscript.

→ We would like to inform you that meta-diamide stand for the chemical class of insecticide. For example, Broflanilide is called a meta-diamide insecticide. We think that it is better to use ‘meta-diamide’ itself to highlight in the chemical class of our compounds.

In the abstract, the term “good levels of physicochemical properties” is too general and meaningless. Why not denominate that water solubility and log P values were measured.

→ corrected

Regarding the physicochemical and biological effects of fluorine and polyfluorinated groups, I like to draw the attention of authors on three recent reviews by Muller, by Pertusiati et al. and by Ojima et al. published in the book “Fluorine in Life Sciences”, Academic Press, 2019, pp 91-139. Pp 141-180 and pp 181-211. This should also be considered at page 4, last paragraph and in ref. [31].

→ We would like to add the exact references what you mentioned. Actually, it was not easy to fine the exact references. Could you please address the each reference to us?

Page 2: it is not clear whether compounds 1b-1d are known compounds described in the patent referenced in [24] already.

→ For reference 24, this is for the condesation reaction. For 1b, it is already known. Please see the Material and Methods section. We added references for 1b. In the cases of 1c and 1d, it is not reported any publication yet. In Material and Methods section, we addressed the analytic data as much as we can.

Finally, authors might want to comment on whether the SF5 derivatives might compete with the CF(CF3)2 lead in terms of economy and persistence in nature.

→ YES, thank you very much for your comment.

Round 2

Reviewer 4 Report

Dear authors,

I guess that most of the readers of Molecules, like myself, are not working in pesticide chemistry. For those colleagues, the authors might like to give a short word of explanation of the term "meta-diamide insecticide".

Here are the exact bibliographic data the authors requested. Each of the reviews are chapter of the book Fluorine in Life Sciences. Pharmaceuticals, Medicinal Diagnostics, and Agrochemicals, Eds., G. Haufe, F. Leroux, Academic Press, San Diego, 2019.

Chapter 2: K. Müller, Fluorination patterns in small alkyl groups: their impact on propierties relevant to drug discovery, pp 91-139.

Chapter 3: F. Pertusati, M. Serpi, E. Pileggi, Polyfluorinated scaffolds in drug discovery, pp 141-180.

Chapter 4: L. Xing, T. Honda, L. Fitz, I. Ojima, Case studies of fluorine in drug disvovery, pp 181-211.

Besides the mentioned chapters, also chapter 1 of the above mentioned book might be interesting for the authors: K. D. Dykstra, N. Ichiishi, S. W. Krska, P. F. Richardson, Emerging fluorination methods in organic chemistry relevant for life science applications, pp 1-90.

Do authors have information whether the SF5 derivatives might compete with the CF(CF3)2 lead in terms of economy and persistence in nature?

Author Response

Thank you very much for your comments and suggestions. It would be very helpful for our research in the future.

I would like to address in most points.

I guess that most of the readers of Molecules, like myself, are not working in pesticide chemistry. For those colleagues, the authors might like to give a short word of explanation of the term "meta-diamide insecticide".

→ Corrected

→ For the exact bibliographic data you addressed, we added in the reference section from 31 to 34.

Do authors have information whether the SF5 derivatives might compete with the CF(CF3)2 lead in terms of economy and persistence in nature?

→ In the case of CF(CF3)2 and SF5, we would answer that it has not been obtained any information yet.

Sincerely,

Seong Jun Park
